# Manipulation of the Phytochemical Profile of Tenderstem^®^ Broccoli Florets by Short Duration, Pre-Harvest LED Lighting

**DOI:** 10.3390/molecules27103224

**Published:** 2022-05-18

**Authors:** Faye M. A. Langston, James M. Monaghan, Olivia Cousins, Geoffrey R. Nash, John R. Bows, Gemma Chope

**Affiliations:** 1Natural Sciences, Streatham Campus, University of Exeter, Exeter EX4 4PY, UK; g.r.nash@exeter.ac.uk; 2Fresh Produce Research Centre, Harper Adams University, Edgmond TF10 8JB, UK; jmonaghan@harper-adams.ac.uk (J.M.M.); ocousins@harper-adams.ac.uk (O.C.); 3PepsiCo R&D, Leicester LE4 1ET, UK; john.bows@pepsico.com (J.R.B.); gemma.chope@pepsico.com (G.C.)

**Keywords:** glucosinolates, light-emitting diodes (LEDs), light wavelength, carotenoids, brassica, phenolics, controlled environment farming

## Abstract

Light quality has been reported to influence the phytochemical profile of broccoli sprouts/microgreens; however, few studies have researched the influence on mature broccoli. This is the first study to investigate how exposing a mature glasshouse grown vegetable brassica, Tenderstem^®^ broccoli, to different light wavelengths before harvest influences the phytochemical content. Sixty broccoli plants were grown in a controlled environment glasshouse under ambient light until axial meristems reached >1 cm diameter, whereupon a third were placed under either green/red/far-red LED, blue LED, or remained in the original compartment. Primary and secondary spears were harvested after one and three weeks, respectively. Plant morphology, glucosinolate, carotenoid, tocopherol, and total polyphenol content were determined for each sample. Exposure to green/red/far-red light increased the total polyphenol content by up to 13% and maintained a comparable total glucosinolate content to the control. Blue light increased three of the four indole glucosinolates studied. The effect of light treatments on carotenoid and tocopherol content was inconclusive due to inconsistencies between trials, indicating that they are more sensitive to other environmental factors. These results have shown that by carefully selecting the wavelength, the nutritional content of mature broccoli prior to harvest could be manipulated according to demand.

## 1. Introduction

Broccoli (*Brassica oleracea*) is recognised as being a particularly nutritious vegetable, often being labelled as a ‘superfood’. As well as containing a number of essential micronutrients, broccoli also contains potential health-promoting phytochemicals. These comprise glucosinolates (GLS), phenolic compounds, tocopherols, and carotenoids. GLS are secondary plant metabolites mainly found in vegetables of the Brassicaceae family, including broccoli. There is an increasing body of research that supports their chemopreventive action, especially sulforaphane, the metabolite and bioactive form of the GLS glucoraphanin [1]. Furthermore, broccoli is a particularly high source of β-carotene, contributing to vitamin A intake as well as the carotenoid lutein.

Phytochemicals have a protective role in plants and are produced to aid in fungi and pest resistance and to help defend against environmental stressors, such as ultraviolet radiation and temperature extremes. Varying pre-harvest and post-harvest conditions can, therefore, affect a plant’s phytochemical profile [2]. Light, as the energy source for photosynthesis, is the primary environmental factor impacting plant growth and development. Exposure of plants to red, blue, and UV light has been reported to influence phytochemical production, with the use of light-emitting diodes (LEDs) allowing for the selection of specific wavelengths and intensities to optimise accumulation of plant antioxidant compounds [3]. Short-term pre-harvest red light increased lutein and β-carotene in basil, red pak choi and tatsoi, as well as total phenols in red pak choi, parsley, tatsoi, and basil, but not in beet or mustard. Exposure to blue light has been reported to have a beneficial effect on carotenoid, total phenolics [4] and GLS [5,6] in a number of plants, alongside increasing biomass. Although there are some commonalities in responses between plants, an increasing number of studies have shown that the response of plants to light quality is species and cultivar specific [7,8].

Controlled environment farming provides the opportunity to modify, control, and monitor many aspects of plant growth environments, including acute adjustment of light parameters. It enables a shorter delivery distance and time period between harvest and consumer, reducing the chance or degree of crop spoilage and deterioration and increasing the nutritional value of the product received by the consumer.

The market for broccoli continues to grow, with different varieties now being widely available, such as purple sprouting broccoli, Belleverde^®^, and broccoli microgreens. Tenderstem^®^ is a commercially available kalian (*Brassica oleracea*, Alboglabra group) and conventional (*Brassica oleracea*, Italica group) broccoli hybrid. It is characterised by a long slender stalk, tender from stem to floret, with a milder sweeter texture than traditional broccoli. Several studies have investigated the impact of exposing broccoli to different wavelengths of light during growth; however, most of these studies have used broccoli sprouts/microgreens (*Brassica oleacea var. italica*) [9,10,11] rather than mature broccoli and none have investigated the impacts on Tenderstem^®^ broccoli. As a plant light-mediated phytochemical response is thought to be species and cultivar dependent, it is important to capture the individual response of the cultivar to determine its optimal growth conditions.

This work investigates the effect of exposing Tenderstem^®^ broccoli to different wavelengths of light in the final growth stages before harvest on the inherent GLS, carotenoid, tocopherol, and total polyphenol content (TPC). Plants were grown in a greenhouse under ambient sunlight with supplementary LED lighting until harvest (Co), or transferred prior to harvest to either a blue LED light treatment (B), or green, red, far-red, i.e., blue light excluded, LED light treatment (G-R-FR).

## 2. Results

### 2.1. Effect of Light Treatment on Tenderstem^®^ Morphology

Fresh shoot weight (FSW) was recorded upon harvest (Table 1). The greatest FSW was observed in control trial 1 harvest 1 (T1H1), which was substantially greater than the FSW of the other light treatments in this harvest. Across all harvests, the FSW under blue light was lower than that of the same harvest. Other than in trial 2 harvest 2 (T2H2), G-R-FR light appears to have reduced FSW; however, the large variation between samples makes it difficult to draw any robust conclusions. Additionally, the height from the base of the plant (aboveground) to the spike was measured before each harvest (Table 2). The G-R-FR and blue LED racks allowed for a maximum height of 80 cm; therefore, as in the second harvest in both trials, the G-R-FR and blue plants were trimmed to 80 cm.

### 2.2. Effect of Light Treatments on Phytochemical Content of Broccoli

#### 2.2.1. Total Polyphenols

The TPC of the Tenderstem^®^ broccoli florets was determined after each harvest from each light treatment (Figure 1). Regardless of light treatment, trial 1 harvest 2 (T1H2) contained the highest values of TPC. Overall, G-R-FR light optimises TPC in Tenderstem^®^ broccoli, obtaining high quantities in both trials, whereas (with exception of T2H2), the TPC of broccoli under blue light was not significantly different to the control.

#### 2.2.2. Glucosinolate Content

The total and individual GLS contents were measured in all the broccoli samples (Figure 2). Trial 2 harvest 1 (T2H1) contained the highest total GLS regardless of light treatment, due to the contribution of significantly higher levels of 4-methoxyglucobrassicin, glucobrassicin, and glucoraphanin (*p* < 0.05). In contrast, T2H2 contained the lowest total GLS by a significant amount in both the control and G-R-FR light treatments (*p* < 0.05). With increased exposure to blue light, the total GLS content decreases in broccoli. The total GLS obtained under G-R-FR light, with the exception of T2H1, is comparable to the content obtained in the control broccoli.

4-methoxyglucobrassicin and glucoraphanin follow a similar trend between trials, whereby there is a small decrease (with the exception of the control glucoraphanin) between T1 harvests and a great decrease between T2 harvests. However, although overall the wavelength of the light does not appear to significantly (*p* < 0.05) impact 4-methoxyglucobrassicin content, G-R-FR and blue light decreased the Tenderstem^®^ glucoraphanin. In contrast, glucoiberin content increases with extended exposure to G-R-FR and blue light, increasing significantly after the two weeks of extra light treatment in both trials under G-R-FR light and T1 under blue light (*p* < 0.05). The highest value was obtained in T2H2 under G-R-FR light (19.9 mg/100 dw). This is distinctly different from the control, where the glucoiberin content decreases across these two weeks. Therefore, it appears that the plant initially responds to the light change by decreasing glucoiberin content, as observed when comparing blue and G-R-FR to the control in the first harvest. However, after prolonged exposure, the levels increase again.

The results of glucobrassicin across all the harvests follow very different trends in T1 to T2, inferring that other environmental factors have a greater influence over glucobrassicin concentrations than the change in light wavelength. In T1, glucobrassicin does not change between harvests or light condition. However, in T2, there is a significant drop between harvests, with a reduction of 72%, 60%, and 24% in the control, blue, and G-R-FR light, respectively. Similarly, for neoglucobrassicin, there is a significant drop between harvests in both trials (with exception of G-R-FR in T1). The greatest drop is observed in the control, where in both trials neoglucobrassicin decreases by 5.3–5.8 mg/100 g dw. With exception of blue T2H2, exposure to G-R-FR and blue light significantly raised the levels of 4-hydroxyglucobrassicin (*p* < 0.05) compared to the control, of which the highest level (37.92 mg/100 g dw) was detected under G-R-FR light T1H2.

#### 2.2.3. Carotenoids and γ-Tocopherol

β-carotene, lutein, and γ-tocopherol were determined in the Tenderstem^®^ florets following harvests (Figure 3). α-Tocopherol was not detected in any of the samples. The blue LED light had a potential positive impact on β-carotene in T1, but had no significant impact in T2, with plants obtaining levels similar to the control. G-R-FR light to some extent reduced β-carotene and lutein content in T1, but this effect was not exacerbated with prolonged exposure nor observed in T2. Interestingly, the same trend as β-carotene is observed in lutein and γ-tocopherol, where in T1 there is a great increase in the carotenoids over the 2 weeks, but in T2 they significantly decrease between harvests. γ-Tocopherol dramatically soars between harvests in T1 for plants exposed to the control, G-R-FR, and blue light treatments, increasing by 460%, 331%, and 592%, respectively. In contrast, in T2, the levels are drastically lower in both harvests with levels not exceeding 7.77 mg/100 g dw, in comparison to the lowest level of 29.74 mg/100 g dw, as observed in T1.

## 3. Discussion

When discussing the results of this study, it is important to note that the movement of the plants to the new LED light treatments was accompanied by a change in light intensity. LED treatments had constant levels of irradiance of 160 μmol PPFD over 18 h, whereas the control crop had greater but variable ambient light levels, peaking at ~500 μmol PPFD. Light intensity, as well as wavelength, has a crop dependent effect on phytochemical production [12]. A negative correlation between carotenoid production and increasing light intensity has been observed previously in broccoli microgreens [13]. The production of polyphenols, however, has been observed to increase with increasing light intensity in broccoli [13], tatsoi, mustard, red pak choi and kohlrabi [14] microgreens. However, most of the work on the influence of light intensity on *Brassica* plant phytochemicals has been conducted on microgreens; therefore, it is difficult to extrapolate these findings to mature plants.

G-R-FR light has a slight beneficial effect on the accumulation of polyphenols, with significantly higher TPC in T1H1 (*p* < 0.05) and similar results across the other harvests compared to the control. The levels of TPC in broccoli grown under blue light are comparable to the control in all harvests except T2H2, where it has significantly (*p* < 0.05) lower TPC. However, as TPC increases with light intensity and the control frequently experiences higher light intensities than the LED treatments, the effects of different wavelengths of light may have been masked by the effect of light intensity. An increase in TPC following the exposure to a high proportion of red/far-red light preharvest and post-harvest has been observed in other broccoli studies [15,16,17]. The biosynthesis of polyphenols has been shown to respond to different lighting conditions, partly due to particular wavelengths having the ability to activate the phenylalanine ammonia-lyase (PAL) gene [18]. PAL is the enzyme that catalyses the first step of the biosynthesis of phenylpropanoids. Consequential enzymatic reactions result in the production of polyphenols derived from the phenylpropane C_6_-C_3_ structure. Polyphenols protect plants against ultraviolet radiation; therefore, it is not unexpected that when plants have been exposed to light in the UV band or wavelengths close to this band, an increase in the biosynthesis of polyphenols has been observed. However, these effects have been shown to be species or cultivar specific [19].

Our study found that growing broccoli under blue and G-R-FR light in the final weeks before harvest somewhat negatively impacts the total GLS content. Increased exposure to blue light decreased the total GLS content by 23% and 49% in T1 and T2, respectively. Broccoli grown under G-R-FR and blue light in T1H2 and under G-R-FR light in T2H1 obtained significantly lower total GLS than the control; the other harvests were comparable to the control. These results were similar to those by Steindal et al. (2016) [15], who found that red, far-red, red and far-red or blue supplemental LED light treatment either decreased or had no significant impact on mature broccoli total GLS.

Despite the total GLS content being slightly greater overall in the control treatment, some of the individual indole GLS were significantly increased by the other light treatments, which were as follows: 4-hydroxyglucobrassicin, neoglucobrassicin, and 4-methoxyglycobrassicin by blue light and 4-hydroxyglucobrassicin and neoglucobrassicin in G-R-FR light. Similarly, Kopsell et al. (2014) [10] reported the indole GLS to be the most responsive to the increase in blue light ratio in broccoli microgreens. Conversely, the relationship between blue light and indole GLS was not observed in the mature broccoli floret study by Steindal et al. (2016) [15]. GLS biosynthesis in broccoli responds dynamically in reaction to temperature changes. The total indole GLS content increased by 24% at 18 °C compared to broccoli grown at 12 °C [20], whereas the total aliphatic GLS content in broccoli sprouts were 45% and 125% greater at 11 °C and 33 °C, respectively, than broccoli sprouts at 22 °C [21]. These observations were further supported by Pereira et al. [22], who found that indole production was greatest in broccoli sprouts grown in intermediate temperatures and aliphatic production greatest at the extremes. In the study by Steindal et al. (2016) [15], the broccoli florets were grown at either 12 °C or 15 °C after germination, in contrast to our study, which maintained temperatures of 19–20 °C and research by Kopsell et al. (2013, 2014) [10,11] performed at 23 °C or 24 °C. Therefore, the temperature difference could explain the difference between this study and the work by Steindal et al. (2016) [15].

Although blue light did cause an increase in indole compounds, it was at the expense of aliphatic compounds, thus, having no overall effect on total GLS content. There is an established antagonistic relationship between the indole and aliphatic clades [23]. Glucoraphanin decreased with increased exposure to the G-R-FR and blue light treatments. One week of blue LED illumination had little effect on the compound; however, by week three, the levels had significantly decreased. Under G-R-FR illumination, the impact was significant after the first week’s exposure, decreasing further by the third week. In comparison, glucoiberin content decreased initially but adjusted by the third week, bringing its content to a comparable level to the control. Across all light treatments, indole compounds comprised 24–33% of the total GLS and aliphatic 67–76%. Therefore, if the aim is to increase indole GLS, altering plant environment parameters, i.e., light quality at intermediate temperatures, is advisable. However, if the aim is to raise aliphatic GLS or total GLS (as the aliphatic compounds comprise the majority of the GLS), temperature extremes should be optimised for plant growth.

Correlation coefficients were calculated to determine significant trends between the GLS. The strongest relationship was found between glucoraphanin, glucobrassicin, and 4-methoxyglucobrassin, exhibiting a significant positive correlation across G-R-FR, blue, and control treatments. The relationship between these compounds has been observed before in studies investigating light quality on broccoli sprouts [10,24], and this correlation would suggest that the upregulation of glucoraphanin under light treatments is concurrent with the upregulation of glucobrassicin and 4-methoxyglucobrassicin. A greater sample size than the one used here (16) would allow these correlations to be explored more fully. In T1, exposure to blue LED light had a slight positive impact on β-carotene; however, this trend was not replicated in T2. G-R-FR light negatively affected the β-carotene content, obtaining significantly lower values in all the harvests, except T2H1 (*p* < 0.05). Across the harvests, the control broccoli had, on average, 11% and 16% higher levels of lutein than blue and G-R-FR light, respectively. Carotenoids are accessory light-harvesting pigments that absorb wavelengths between 400–500 nm and also act as photoprotective agents. Genes that regulate carotenoid biosynthesis pathways have been shown to be partially influenced by light. However, the transcriptional regulation of carotenoid biosynthetic genes varies between species [25]; therefore, different species and cultivars respond independently to changes in light quality [7,8]. The results of this current study are consistent with the findings of Kopsell et al. (2013, 2014) [10,11], where lutein and β-carotene content in sprouting broccoli responded similarly to light treatments. In an earlier study, they found that short-duration blue light slightly increased β-carotene accumulation in broccoli shoot tissue, in contrast to lutein where there was no significant difference between treatments.

Unlike carotenoids and phenolics that directly absorb light, tocopherols are indirectly influenced by changes to light through subsequent changes to their metabolic pathways. Generally, tocopherols have been shown to increase with exposure to red light [26]. However, in our study, it appears that other environmental or developmental factors are more dominant in affecting γ-tocopherol content than light wavelength. This is suggested by the similar dramatic rise observed between T1H1 and T1H2 across all the light treatments. The profile and total content of tocopherols are highly sensitive to changes in environmental stresses [27], increasing under certain conditions, i.e., high light intensity, water deficit and low temperature. γ-tocopherol is particularly sensitive to low water, increasing in *Arabidopsis* by 13.5 fold in response to water stress [28]. Therefore, the dramatic increase observed between T1H1 and T1H2 could be due to the plants experiencing other external stressors, such as a water deficit.

## 4. Materials and Methods

### 4.1. Plant Experimental Set-Up

#### 4.1.1. Plant Growth before Treatment

The plants were grown in a glasshouse bay at Harper Adams University, Shropshire, UK (Grid Ref SJ 711200). The average glasshouse conditions were 18 °C, 62% relative humidity for Trial 1 (T1) and 19 °C, 77% relative humidity for Trial 2 (T2). Supplemental lighting (Phytolux, Worcester, UK) at 94 µmol m^–2^ s^–1^ PPFD was provided with a 16/8 h day/night photoperiod. T1 was sown on 23 February 2021; T2 was sown on 19 April 2021. For each trial, 120 seeds were sown into Levington M2 compost (ICL Ltd., Ipswich, UK), in 5 × 8 cell modular trays (Plant Pak P40, Desch Plantpak Ltd., Maldon, UK). Three weeks after sowing, 80 seedlings of uniform size were transplanted into 1 L pots (114 mm height × 130 mm diameter) of M2 compost. After a further 3 weeks, 60 plants of uniform size were transplanted into 4.5 L pots (185 mm height × 190 mm diameter) of M2 compost. The plants were then fed weekly with Vitax Vitafeed (Vitax Ltd., Leicester, UK), providing NPK (1:1:1) diluted 1:200 (after Wurr et al. (2002) [29]).

#### 4.1.2. Light Treatments

LED lighting treatments were applied using four fully enclosed vertical growing racks in a compartment adjacent to the control compartment, with two shelves distributed evenly per rack. Each rack had 2 1188 mm long LED lights (Horti-blade, The Vexica Group Ltd., Leeds, UK) fitted horizontally at 80 cm above the top of the pots (see Figure 4). One shelf was fitted with blue LED lights, the other shelf was fitted with four-channel blue, white, red, far-red LED lights. The following three light treatments were implemented: (1) Green, Red, Far-Red (G-R-FR) (400–780 nm; ~160 µmol PPFD); (2) Blue (B) (400–499 nm; ~160 µmol PPFD); (3) Control (Co), where plants remained in the original compartment with ambient sunlight and supplemental lighting. The blue LED were on the top shelf of two of the four racks and the G-R-FR LED were on the top shelf of the other two racks. The spectra of the LED lighting treatments, supplementary LED and sunlight within the pre-treatment glasshouse bay were measured using a spectroradiometer (Asensetek Lighting Passport, Allied Scientific Pro, QC, Canada) (Table 3).

The photoperiod for LED lighting treatments and supplemental lighting was 16 h from 06:00 to 22:00. The photosynthetic photon flux density (PPFD) of the LED growing chambers was measured at ~20 cm above the canopy of the plants at the start of LED light treatments, at ~160 μmol m^–2^ s^–1^ PPFD for both B and G-R-FR treatments. The plants in the control compartment received ambient sunlight with supplemental LED lighting (Phytolux, Worcester, UK) at 94 μmol m^–2^ s^–1^ (Table 4).

The average temperature and relative humidity during the pre-lighting treatments and lighting treatments are summarised in Table 5.

#### 4.1.3. Timing of Treatments

Plant development was monitored weekly. The majority of plants had a terminal meristem of >1 cm in diameter in week 10 in T1 and week 9 in T2 with 12–13 leaves on the main stem. The lighting treatments started when the majority of plants had axial meristems in the top two leaf axes of >1 cm diameter the following week (week 11, T1; week 10, T2). Twenty plants were placed under B LEDs, twenty were placed under G-R-FR LED with five plants per shelf and twenty plants remained in the original compartment, blocked into groups of five plants.

#### 4.1.4. Floret Samples

The plants were grown for one week after starting the light treatments before the first central, primary floret (Figure 4b) was sampled (week 12, T1; week 11, T2). The height of the plant from base to top of the floret was recorded, the floret was cut above the last pair of leaves, and fresh weight was recorded. The plants remained under light treatments for another two weeks, when the secondary florets were harvested as described with the primary floret (week 14, T1; week 13, T2). Therefore, in total, two samples from each plant in each light treatment were taken for phytochemical analysis (2 × 20 × 3) and this was repeated for trial 2. In addition, the whole plant biomass was measured. SPAD readings were also taken the day before the final harvest using a SPAD meter (Chlorophyll Meter SPAD-502Plus, Konica Minolta Sensing Europe B.V, Warrington, UK).

### 4.2. Determination of Total Polyphenol Content

TPC was determined using the Folin–Ciocalteu assay. The samples were quantified using a gallic acid standard curve, and the final reported contents were given as gallic acid equivalents (GAE). The samples (200 mg) were accurately weighed into 15 mL capacity tubes. A 10 mL sample of pre-heated 70% methanol (60 °C) was then added, and the samples were agitated for 2 min. The samples were then heated at 60 °C for 2 h, agitated and then cooled, shaken, and centrifuged at 3000 rpm for 10 min. A portion of the supernatant (250 µL) was taken and Folin–Ciocalteu reagent (250 µL) was added, along with 1 mL of 10% sodium carbonate solution and 8 mL of water. The sample was mixed and allowed to stand for 90 min before measurement at 760 nm. The samples were prepared alongside standard solutions of gallic acid in 70% methanol at 50–2000 µg mL^−1^.

### 4.3. Determination of GLS Content

The broccoli samples were freeze-dried and powdered (Nutribullet 600 W Personal Blender, Los Angeles, CA, USA) to homogenise the material prior to analysis. Broccoli GLS was measured using a method that converts the GLS to the equivalent desulfo-GLS, as described previously [30].

### 4.4. Determination of Carotenoids and Tocopherol

The freeze-dried samples (0.5 g) were homogenised with 1:1 tetrahydrofuran/methanol (*v/v*; 20 mL), 0.1 g sodium carbonate and 25 μL of internal standard (echinenone; 1 mg/mL in dichloromethane). The resulting suspension was filtered and washed with 1:1 tetrahydrofuran/methanol (20 mL). The combined tetrahydrofuran/methanol filtrates were transferred to a large separating funnel and washed with petroleum ether (40–50 fraction, containing 0.1% BHT; 20 mL) and 10% NaCl solution (20 mL), drawing off the lower tetrahydrofuran/methanol/aqueous phase to waste each time. The combined petroleum ether fraction was washed with water (3 × 500 mL), drawing off the lower aqueous wash each time to waste. The petroleum ether layer was collected and transferred to a smaller separating funnel, separated into a 250 mL round-bottomed flask, and evaporated to near dryness at 40 °C (pressure of 150 bar) in a rotary evaporator. Petroleum ether (10 mL) was added to redissolve the residue, which was transferred to a 25 mL round-bottomed flask and the sample evaporated to dryness. The residue was redissolved in 5 mL of dichloromethane, transferred to a volumetric flask, and made up to 20 mL with the mobile phase (acetonitrile containing 0.1% formic acid). An aliquot (300 μL) was transferred to an amber autosampler vial for HPLC analysis. The samples were analysed using HPLC-UV. An isocratic method, with a solvent of 65% acetonitrile and 35% methanol containing 0.016% tributylamine, was used to elute the carotenoids and tocopherols. UV detection at a wavelength of 450 nm for carotenoids, and 290 nm for tocopherols were used. A Waters ODS2 C18 column (250 × 4.6 mm, 5 µm particle size) was used with an oven temperature of 35 °C (flow rate: 1.5 mL min^−1^, injection volume: 7.5 μL). Under the conditions applied, beta-carotene eluted at approximately 26.9 min, lutein at 4.3 min, zeaxanthin at 4.5 min, alpha-tocopherol at 9.5 min, and beta- and gamma-tocopherol co-eluting at 8.3 min. Quantification was undertaken using calibration curves of the peak area versus concentration of freshly prepared standard solutions of the carotenoids and tocopherols.

### 4.5. Statistical Analysis

Significant differences (*p* < 0.05) were determined by Excel software using one-way analysis of variance (ANOVA) and Tukey’s Test.

## 5. Conclusions

In summary, this study has shown that the phytochemical profile of Tenderstem broccoli can be manipulated through altering light wavelength conditions prior to harvest. Depending on the phytochemical of interest, careful selection of other environment parameters is integral, e.g., temperature when assessing glucosinolates, and water availability for tocopherols. Numerous studies have focused on the growth of a range of leafy crops, including salad leaves, microgreens and herbs [3,4,5,6,7,8,9,10,11]. This study is the first to quantify the manipulation of phytochemical levels in a mature glasshouse grown vegetable brassica, using a period of LED lighting prior to harvest. More research with larger numbers of replicates is required in mature broccoli to develop a fuller understanding of the effect of different wavelengths of light on the phytochemical profiles. Furthermore, measuring energy consumption during plant growth would be important for economical consideration in commercial application. Additional research in this field could enable the delivery of broccoli with higher contents of specified compounds without the need for extensive breeding programmes.

## Figures and Tables

**Figure 1 molecules-27-03224-f001:**
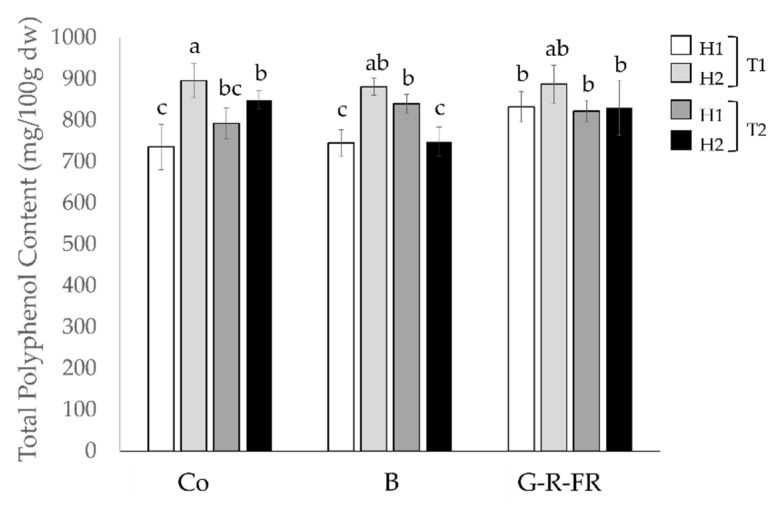
Total polyphenol content of broccoli samples measured in dry weight across all harvests and light treatments. H in the legend relates to harvest number and T to trial number. Co (control), B (blue), G-R-FR (green/red/far-red). The different letters represent significant differences between samples (*p* < 0.05).

**Figure 2 molecules-27-03224-f002:**
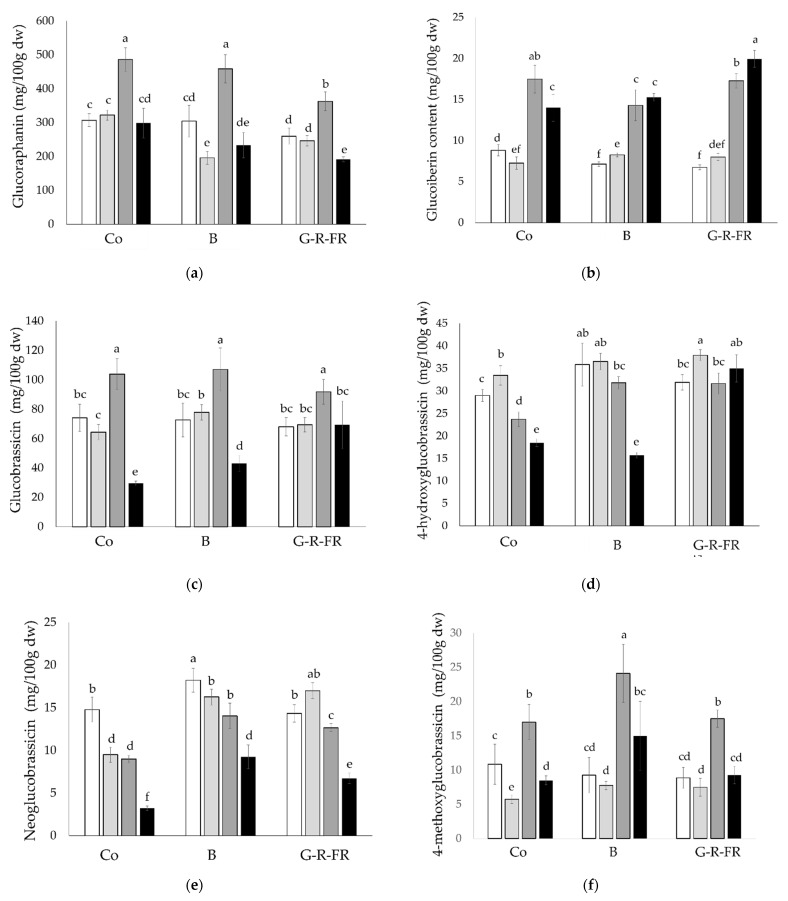
Individual glucosinolate, (**a**) glucoraphanin, (**b**) glucoiberin, (**c**) glucobrassicin, (**d**) 4-hydroxyglucobrassicin, (**e**) neoglucobrassicin, (**f**) 4-methoxyglucobrassicin, and (**g**) total glucosinolate content of broccoli samples across all the harvests and light treatments. The different letters represent significant differences between samples (*p* < 0.05).

**Figure 3 molecules-27-03224-f003:**
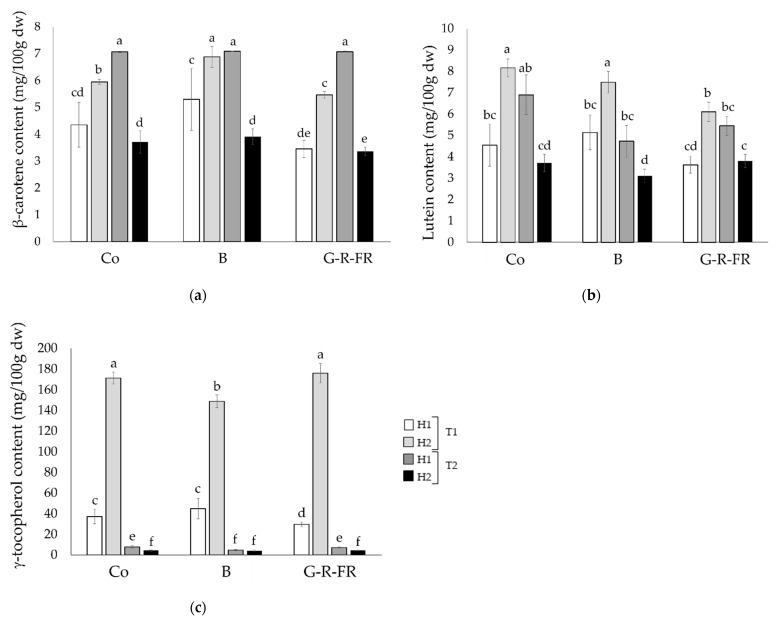
(**a**) Levels of β-carotene, (**b**) lutein, and (**c**) γ-tocopherol were measured in broccoli samples across all harvests and light treatments. The different letters represent significant differences between samples (*p* < 0.05).

**Figure 4 molecules-27-03224-f004:**
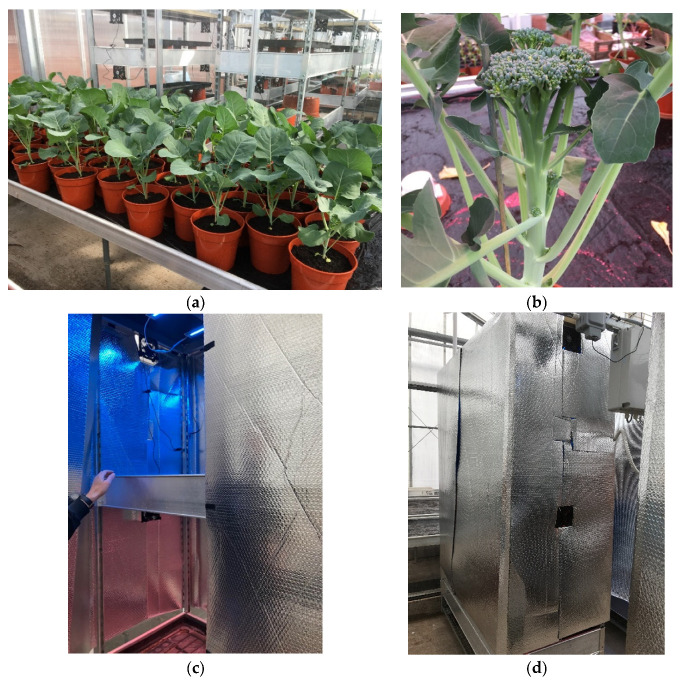
Images of the plants and lighting conditions. (**a**) Plants at week 8, trial 1, 19 April 2021, (**b**) central primary floret, week 12, trial 1, 13 May 2021, (**c**) LED lighting rack with blue LED lights on the top shelf and four-channel blue, white, red, far-red LED lights on the bottom; (**d**) LED shelving container when closed.

**Table 1 molecules-27-03224-t001:** Tenderstem^®^ fresh shoot weight (g) mean and standard deviation at each harvest. Co (control), B (blue), G-R-FR (green/red/far-red).

	Trial 1						Trial 2					
	Harvest 1			Harvest 2			Harvest 1			Harvest 2		
	M	SD	SE	M	SD	SE	M	SD	SE	M	SD	SE
Light treatment	(Fresh weight g)											
Co	110.23	20.45	4.57	71.45	13.38	2.99	53.58	21.73	4.86	53.22	11.16	2.50
B	84.53	19.29	4.31	59.78	14.48	3.24	33.74	19.67	4.40	39.22	18.46	4.18
G-R-FR	89.09	19.88	4.44	68.73	19.21	4.30	45.32	15.37	3.44	63.81	21.77	4.87

**Table 2 molecules-27-03224-t002:** Tenderstem^®^ height from the plant base, aboveground, to spike (cm) mean and standard deviation at each harvest. Co (control), B (blue), G-R-FR (green/red/far-red).

	Trial 1						Trial 2					
	Harvest 1			Harvest 2			Harvest 1			Harvest 2		
	M	SD	SE	M	SD	SE	M	SD	SE	M	SD	SE
Light treatments	(Height to spike cm)											
Co	72.60	6.77 4.17	1.51	95.75	9.62 0.00	2.15	80.55	5.16 4.29	1.15	114.75	10.45 0.00	2.34
B	70.25	4.17	0.93	80.00	0.00	0.00	77.95	4.29	0.96	80.00	0.00	0.00
G-R-FR	70.75	3.13	0.70	80.00	0.00	0.00	78.75	2.22	0.50	80.00	0.00	0.00

**Table 3 molecules-27-03224-t003:** The fraction of spectrum distribution of the LED light sources. Indicative solar light level recorded at 13:00 (solar midday) 5 July 2021 above the control treatment plants. Co (control), B (blue); G-R-FR (green/red/far-red).

Light Source	Distribution (%)				
	380–399 nm UV	400–499 nm Blue	500–599 nm Green	600–700 nm Red	701–780 nm Far-Red
B LED	0	99	1	0	0
G-R-FR LED	0	2	11	62	25
Co LED	0	7	11	81	1
Sunlight	0	19	28	30	23

**Table 4 molecules-27-03224-t004:** The total photon flux density (PFD) and photosynthetic photon flux density (PPFD) of the LED light sources. Indicative maximum solar light level recorded at 13:00 (solar midday) 5 July 2021 above control treatment plants.

Light Source	Irradiance (μmol m^–2^ s^–1^)	
	PFD 380–780 nm	PPFD 400–700 nm
B LED	160	160
G-R-FR LED	213	160
Co LED	95	94
Sunlight	675	517

**Table 5 molecules-27-03224-t005:** Average temperature and relative humidity during light treatments. Co (control), B (blue), G-R-FR (green/red/far-red).

Trial	Light Treatment	Growth Period	Temperature °C			Relative Humidity (%)		
			Average	Max	Min	Average	Max	Min
1	G-R-FR	Week 11–13	22.5	43.8	10.9	78.4	100.0	14.7
	B	Week 11–13	20.2	32.5	11.0	79.1	100.0	41.1
	Co	Week 1–10	17.5	39.8	4.3	64.1	100.0	22.2
	Co	Week 11–13	20.0	36.2	9.3	75.9	100.0	38.4
2	G-R-FR	Week 11–13	25.2	40.8	16.3	72.3	91.6	44.2
	B	Week 11–13	24.6	37.3	16.6	74.0	91.4	49.6
	Co	Week 1–10	22.0	45.8	5.4	67.7	100.0	11.6
	Co	Week 11–13	24.5	41.0	16.1	61.7	83.5	25.7

## Data Availability

Not applicable.

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
