# Peer review of "Manipulation of the Phytochemical Profile of Tenderstem^®^ Broccoli Florets by Short Duration, Pre-Harvest LED Lighting"

_molecules, 2022, doi:10.3390/molecules27103224_

Round 1

Reviewer 1 Report

This paper contains interesting studies on Effect of pre-harvest LED illumination conditions on the phyto-chemical content of Tenderstem® broccoli florets.  Although many studies have been conducted on that is still lacking in English and informations. In addition, this study is significant for the experimental regions. However, this manuscript still needs improving in writing logic and the analysis of the discussion are not clear. It is recommended to modify   for a better version. Moreover the discussion section is not deep enough, the sentence structure is simple, and there is much repetition.

Title:

- It needs to be modified

Abstract:

-Authors should consider the proposed changes for improving clarity of the contents with focusing good findings.

Keywords:should be change

Abbreviations: should arrange 

Introduction:

-Introduction part is appropriate but few things needed for further improvements especially study hypothesis should be added for the last 5 years.

-Add some studies about the study with highlighting research gaps which necessitated conducting this trial.

Materials and methods:

-this part needs to describe very well . However, it needs more  of the data and information’s for the investigations  regards the  conditions of growth. It is suggested to prepare good data. in the revised version to enhance clarity.

Results and Discussion

-Both parts needs to combine and it needs major revision.

The figures needs more modifying to be more clear and understanding.

In tables, add SE. essential to arrange correlation table…

Conclusion:

-Improve this part with respect to formulated objectives.

References:

-Cross check the references in the text and reference cite. Few references are not as per journal style in the text as well reference section.

Author Response

  1. Title: It needs to be modified. Abstract: Authors should consider the proposed changes for improving clarity of the contents with focusing good findings Keywords: should be change Abbreviations: should arrange

We appreciate thank the reviewer for their comments and for taking the time to read the manuscript. We have modified the title of the paper, so that it reflects the results of the manuscript more concisely. It now reads ‘Manipulation of the phytochemical profile of Tenderstem® broccoli florets by short duration, pre-harvest LED lighting.’ instead of the original title: ‘Effect of pre-harvest LED illumination conditions on the phytochemical content of Tenderstem® broccoli florets’.

We have also removed the numbers from the keywords section and amended the abstract to try its clarity. Manuscripts in this journal don’t require an abbreviation section, and we have defined each abbreviation in the text the first time they appear. We checked to make sure that all terms are properly defined.

  1. Introduction:
  • Introduction part is appropriate but few things needed for further improvements especially study hypothesis should be added for the last 5 years.
  • Add some studies about the study with highlighting research gaps which necessitated conducting this trial.

We thank the reviewer for highlighting this and have added a reference to a paper by Loi et al (reference 3) that provides a comprehensive review of this research area - a review paper that discusses the ‘Challenges and Opportunities of Light-Emitting Diode (LED) as Key to Modulate Antioxidant Compounds in Plants’.

  1. Materials and methods: This part needs to describe very well . However, it needs more  of the data and information’s for the investigations  regards the  conditions of growth. It is suggested to prepare good data. in the revised version to enhance clarity. Results and Discussion:
  • Both parts needs to combine and it needs major revision.
  • The figures needs more modifying to be more clear and understanding.
  • In tables, add SE. essential to arrange correlation table

We thank the reviewer for their feedback regarding  the Methods and Materials and Results and Discussion sections. We have added images (Figure 4, page 9) to the Methods and Materials section, as well as a subsection on the statistical analysis methods used (page 12, lines 386-388). Further clarity over sample numbers and replicated have also been explained to improve further understanding of the study (page 11, lines 339-341) and have also added SE to the tables in the results section as advised. Having made these amendments, we found it was easier to follow the manuscript with separate sections.

  1. Conclusion: Improve this part with respect to formulated objectives.

We have improved the conclusion to address the objectives and include further references to future work (Page 12, lines 399-402).

  1. References: Cross check the references in the text and reference cite. Few references are not as per journal style in the text as well reference section.

We thank the reviewer for pointing this out, and have reviewed and cross checked the references for accuracy and to ensure they match the in-text citations.

Reviewer 2 Report

Line 80,  it should be Table 1

Line 86,  it should be Table 2

Line 270- include the country.

Line 295,  it should be Table 3.1

Line 301,  it should be Table 3.2

Line 303,  it should be Table 3.3

It is necessary to explain the kind of statistical analyses applied.

In my opinion results are inconclusive. The discussion does not explain the variation in results between trials 1 and 2.

Author Response

Point 1:

Line 80,  it should be Table 1

Line 86,  it should be Table 2

Line 270- include the country.

Line 295,  it should be Table 3.1

Line 301,  it should be Table 3.2

Line 303,  it should be Table 3.3

Response 1: The changes to the references of the tables and the inclusion of the country in the description have been made accordingly.

Point 2:

It is necessary to explain the kind of statistical analyses applied.

Response 2: A subsection in the materials and methods section has been added to explain the statistical analysis

Point 3:

In my opinion results are inconclusive. The discussion does not explain the variation in results between trials 1 and 2.

Response 3: The results and discussion section have been reviewed to enable a clearer discussion of the results, as well as addressing the reasoning for the differences in results between the two trials.

Reviewer 3 Report

The manuscript ID molecules-1691320 entitled “Effect of pre-harvest LED illumination conditions on the phytochemical content of Tenderstem® broccoli florets” is an interesting and valuable study. The Authors properly present the current scientific achievements in the field. The aim of the research is clear and well-worded. The conclusions correspond to the results obtained. The manuscript is written in the correct language, its layout and the extracted chapters are logical. The presented content corresponds to the Molecules journal profile. My remarks below:

  1. Please remove numbering from keywords.
  2. Did the authors formulate research hypotheses that they wanted to verify before starting the research? What was the basis and inspiration for the formulation of these hypotheses?
  3. Can the authors take into account the power of the tested lighting sources and energy consumption in the methodology? It would be interesting from an economic point of view.
  4. Please include in the methodology which statistical tests were used to conduct a comparative analysis between the analysed variables.
  5. Please specify in how many repetitions the tests and analyzes have been performed?
  6. Were there any trends and correlations between the analyzed variables?
  7. Is it possible to add photos of test stands, plants, lighting systems to the manuscript?
  8. Manufacturers of lighting systems and analytical equipment should be listed.
  9. In the discussion or introduction, you can supplement the potential of LED lighting related to their low energy consumption and the possibility of emitting specific, selective wavelengths in relation to other species. The use of this type of lighting is particularly interesting in the dynamically developing world-wide cultivation of microalgae intended for energy and dietary purposes or for the production of specific compounds of substances: https://doi.org/10.3390/en13071536, https://doi.org/10.1016/j.biombioe.2016.05.031, https://doi.org/10.3390/antiox10010042

Author Response

  1. Please remove numbering from keywords.

We thank the reviewer for pointing this out, it has been amended accordingly.

  1. Did the authors formulate research hypotheses that they wanted to verify before starting the research? What was the basis and inspiration for the formulation of these hypotheses?

Building on the abundant amount of research on brassica microgreens, including sprouting broccoli, the aim of this study was to see whether these results can be extrapolated to the mature plant which is often used more in commercial manufacturing of food snacks and commodities. From previous research, it was hypothesised that red light would have a positive impact on polyphenol content and that blue light would positively impact aliphatic glucosinolates. However, previous research has that the influence of light quality is species specific and limited research had been conducted on mature broccoli and none on Tenderstem broccoli. Therefore it was an investigation to see the impact on this specific cultivar.

  1. Can the authors take into account the power of the tested lighting sources and energy consumption in the methodology? It would be interesting from an economic point of view.

Unfortunately we do not have the data for this, but agree that this is important from an economic perspective and have therefore mentioned it in the conclusion (Page 12, lines 399-402) for future work.

  1. Please include in the methodology which statistical tests were used to conduct a comparative analysis between the analysed variables.

We agree with the reviewer, and in hindsight realise this should have been included. A subsection in Methods and Materials (page 12, lines 386-388) has been added to address the methods of statistical analyses that have been performed.

  1. Please specify in how many repetitions the tests and analyzes have been performed?

Following the reviewers suggestion we have elaborated on this by including an additional sentence that helps to make this clearer. This can be found in the Methods and Material section under the subheading of ‘4.1.4. Floret samples’ (page 11, lines 339-341).

  1. Were there any trends and correlations between the analyzed variables?

Correlation coefficients were calculated to determine significant trends between the GLS. The strongest relationship was found between glucoraphanin, glucobrassicin, and 4-methoxyglucobrassin, exhibiting a significant positive correlation across G-R-FR, blue, and control treatments. Lutein and β-carotene content were to follow a similar response to each of the light treatments.

  1. Is it possible to add photos of test stands, plants, lighting systems to the manuscript?

We thank the reviewer for this suggestion and agree that adding photos of the experimental set up could improve the understanding of the methodology. Pictures of the lighting systems with the plants have now been inserted into the Materials and Methods section. It can be found in Figure 4, page 9.

  1. Manufacturers of lighting systems and analytical equipment should be listed.

We thank the reviewer for highlighting this. We have made sure that these are included in brackets following their first reference in the methods and materials section as suggested by the journal’s formatting.

  1. In the discussion or introduction, you can supplement the potential of LED lighting related to their low energy consumption and the possibility of emitting specific, selective wavelengths in relation to other species. The use of this type of lighting is particularly interesting in the dynamically developing world-wide cultivation of microalgae intended for energy and dietary purposes or for the production of specific compounds of substances: https://doi.org/10.3390/en13071536https://doi.org/10.1016/j.biombioe.2016.05.031, https://doi.org/10.3390/antiox10010042

We thank the reviewer for bring these papers to our attention. Though insightful and useful, papers 1 & 2 refer to micro algae and we didn’t find them appropriate for this paper. However paper 3 is a good review paper and we have made reference to the third paper in the introduction (reference 3).

Round 2

Reviewer 1 Report

this manuscript still needs improving in writing logic and the analysis of the discussion are not clear. It is recommended to improve for english.

Author Response

Comment 1: This manuscript still needs improving in writing logic and the analysis of the discussion are not clear. It is recommended to improve for English.

We thank the reviewer for reading through the revised manuscript. It has now been read by several native speakers and any punctuation and spelling has been double checked.

Reviewer 2 Report

It is an interesting research which must be studied in greater depth. 

Author Response

Comment 1: It is an interesting research which must be studied in greater depth. 

We thank the reviewer for taking the time to review the revised version of the manuscript and for taking an interest in our research. We have stated page 12, lines 397-402 that further work should be done to develop a fuller understanding of the effect of different wavelengths of light on the phytochemical profiles as well as reviewing potential economic impacts that would influence commercial applications. The manuscript has also been reviewed by native speakers and spelling and punctuation double checked.

Reviewer 3 Report

Thanks a lot  to the Authors for improving the manuscript and taking into account my suggestions and comments. In my opinion, the manuscript can be published in current form.

Author Response

Dear reviewer, thank you very much for taking the time to help us improve our manuscript and for checking the revised version. We highly appreciate your comments and advice!